# Identifying Veterans Using Electronic Health Records in the United Kingdom: A Feasibility Study

**DOI:** 10.3390/healthcare8010001

**Published:** 2019-12-19

**Authors:** Katharine M. Mark, Daniel Leightley, David Pernet, Dominic Murphy, Sharon A.M. Stevelink, Nicola T. Fear

**Affiliations:** 1King’s Centre for Military Health Research, King’s College London, Weston Education Centre, Cutcombe Road, London SE5 9RJ, UKdaniel.leightley@kcl.ac.uk (D.L.); david.pernet@kcl.ac.uk (D.P.); dominic.murphy@combatstress.org.uk (D.M.); nicola.t.fear@kcl.ac.uk (N.T.F.); 2Combat Stress, Tyrwhitt House, Oaklawn Road, Leatherhead KT22 0BX, UK; 3Department of Psychological Medicine, King’s College London, Institute of Psychiatry, Psychology and Neuroscience, De Crespigny Park, London SE5 8AF, UK; 4Academic Department of Military Mental Health, King’s College London, Weston Education Centre, Cutcombe Road, London SE5 9RJ, UK

**Keywords:** electronic health records, mental health, secondary mental health care, national health service, United Kingdom, veterans, feasibility study

## Abstract

There is a lack of quantitative evidence concerning UK (United Kingdom) Armed Forces (AF) veterans who access secondary mental health care services—specialist care often delivered in high intensity therapeutic clinics or hospitals—for their mental health difficulties. The current study aimed to investigate the utility and feasibility of identifying veterans accessing secondary mental health care services using National Health Service (NHS) electronic health records (EHRs) in the UK. Veterans were manually identified using the Clinical Record Interactive Search (CRIS) system—a database holding secondary mental health care EHRs for an NHS Trust in the UK. We systematically and manually searched CRIS for veterans, by applying a military-related key word search strategy to the free-text clinical notes completed by clinicians. Relevant data on veterans’ socio-demographic characteristics, mental disorder diagnoses and treatment pathways through care were extracted for analysis. This study showed that it is feasible, although time consuming, to identify veterans through CRIS. Using the military-related key word search strategy identified 1600 potential veteran records. Following manual review, 693 (43.3%) of these records were verified as “probable” veterans and used for analysis. They had a median age of 74 years (interquartile range (IQR): 53–86); the majority were male (90.8%) and lived alone (38.0%). The most common mental diagnoses overall were depressive disorders (22.9%), followed by alcohol use disorders (10.5%). Differences in care pathways were observed between pre and post national service (NS) era veterans. This feasibility study represents a first step in showing that it is possible to identify veterans through free-text clinical notes. It is also the first to compare veterans from pre and post NS era.

## 1. Introduction

### 1.1. Veterans

The UK’s veteran population, defined by the British government as those who have been in military service for at least one day [1], is estimated to be 2.5 million [2,3]. A small group of veterans experience mental health problems (estimates range from 7% to 22% across psychiatric conditions [4,5,6]), some resulting from their experiences during deployment. Further, veterans show increased rates of psychological difficulties—such as post-traumatic stress disorder, alcohol misuse and suicidal thoughts—compared to civilians [7] and serving military personnel [5].

Recent research suggests that 93% of veterans who reported having a mental health difficulty seek some form of help for their problems, with 86% endorsing informal support through family and friends and 46% endorsing formal non-medical (or welfare) support [8]. To support help-seeking, a range of interventions are currently being developed [9,10,11]. However, it is worth noting that treatment seeking rates vary across studies in the UK. For example, one previous study found that only 25% of veterans sought formal medical support [12] and another, investigating both serving personnel and veterans, reported a rate of 31% [13]. While there has been an apparent increase in the number of military men and women seeking professional help for their mental health difficulties [8], a sizeable minority of this population still do not access appropriate formal medical treatment [8,14]. The longer a presenting mental health problem is left untreated, the more severe and complex it becomes [15], highlighting the importance of veterans in psychological distress seeking help early, before the manifestation of symptoms.

While it is useful to know the profile of those who take the first step in tackling their problem through primary care, we know little about veterans who access secondary mental health care—more specialist care, often delivered in higher intensity therapeutic clinics or hospitals. This is partly because veteran status is not routinely collected for those accessing primary or secondary care [16]. As a result, there is a lack of quantitative information concerning the profile of veterans receiving secondary mental health care and the treatment they receive within the NHS in the UK. Those who access such services, or who are referred to such services from primary care, will, in general, have more complex levels of need compared to those who remain in primary care.

### 1.2. Electronic Health Records

EHR-based systems function as single, complete and integrated electronic versions of traditional paper health records [17]. EHRs have been positioned as a possible “new generation” for mental health research and are now required in the NHS across the UK [18]. The methodological advantages of EHRs—including their longitudinal nature, their size and their detailed coverage of defined populations—make them a salient research asset, providing large numbers of participants and measurement points, as well as the potential for data linkage [19]. EHRs in mental health care provide extremely rich material and analysis of their data can reveal patterns in healthcare provisions, patient profiles and mental and physical health problems [16,20]. This is hugely advantageous for investigating vulnerable subgroups within the wider population.

### 1.3 Current Study

There is currently no flag or indicator for identifying veterans in the UK within EHRs [16,17]. To the best of our knowledge, only one study exists in relation to secondary care and the AF [16]—here, a method was developed to integrate the EHRs of military personnel in England, Scotland and Wales. However, the authors did not attempt to independently identify serving personnel and veterans and utilised an existing military sample to link data from across the UK [16].

In contrast, the current study investigated the utility and feasibility of manually identifying veterans accessing secondary mental health care services through a psychiatric database of a large NHS Foundation Trust in the UK. It is unique in attempting to detect members of the AF who have sought treatment for their mental health difficulties. We discuss utility and feasibility in relation to manually identifying and verifying *probable* veterans from a clinical database and extracting and comparing data related to these veterans.

## 2. Methods

### 2.1 Study Materials

The CRIS system was set up in 2006 as a novel data resource, derived directly from the routine EHRs of the South London and Maudsley (SLaM) NHS Foundation Trust [21]. The data held by CRIS are not representative of the UK. Nevertheless, SLaM is one of Europe’s largest mental health providers, serving over 1.3 million residents in four south London boroughs (Croydon, Lambeth, Lewisham and Southwark) and representing a diverse range of socio-demographic factors [22]. CRIS holds medical records for all secondary mental health care within SLaM, which include all specialist care for hospitalisations, outpatient care, community care, psychiatric liaison services to general hospitals and forensic mental health services.

The CRIS system is composed of: (a) pre-set, structured fields, made up of dates and numeric records—for example, dates of birth and scores on psychological tests such as the Alcohol Use Disorders Identification Test [23]; and (b) open-text, unstructured fields, made up of user-defined text strings—for example, day-to-day clinical notes relating to a particular outpatient psychotherapy appointment. Researchers can search CRIS for any combination of these fields and the system returns relevant records based on the search terms inputted and the fields specified. Pseudo-anonymity is ensured for all records through unique identification numbers, which are separate from NHS numbers; dates of birth that only specify year and month of birth, while day of birth is automatically set to the 1st of the month; and blanked out names (“Mr ZZZZZ”) in the written notes. Results are returned in a spreadsheet format and patient records are updated every 24 h [19].

Ethical approval for the use of CRIS as an anonymised database for secondary analysis was granted by the Oxford Research Ethics Committee (reference 08/H0606/71+5). The current study described here has been approved by the CRIS Patient Data Oversight Committee of the National Institute of Health Research Biomedical Research Centre.

### 2.2. Study Population

This study was focused on veterans who had accessed secondary mental health care services through SLaM. They were identified using a detailed search strategy (see Section 3 for details) following the broad principles of the Cochrane Handbook for Systematic Reviews of Interventions. We included: Veterans who had accessed secondary mental health care services within a ten-year window—1 January 2007 to 31 December 2016. The CRIS register was implemented in 2007, so this was the earliest that EHRs could be accessed. This project commenced in 2018, so 2017 was the last full year that digital records were available.Veterans who had served in the UK AF—we retained records for those whose country of birth was noted as the UK or was left as blank (as we noticed that this field was often left blank if the individual was a UK national).

This study included pre and post national service (NS) era veteran. However, it is important to consider that those who carried out NS may not have voluntarily entered the AF, as it was a legal requirement to serve in the military of the UK between 1949 and 1963. Veterans were determined to be of NS era if they were of legal age to be conscripted into the AF between these dates. It was also hypothesised that there may be differences in care needs older veterans [24].

#### Study Procedure

Identifying: As there is no structured field for flagging (or denoting) probable veterans within CRIS, our objective was to identify this group by applying a military-related keyword search strategy to the unstructured free-text notes filled in for each patient by clinicians. All EHRs identified as potential veteran records were manually checked by a member of the research team (KMM, DL or DP), to ensure they indicated that the patient had served in the military. It must be noted that, although military terms and phrases were used, we were unable to confirm whether patients had served in the AF. In this study, we use the term “probable veteran” from this point onwards. Percentage agreement between KMM, DL and DP was 100% for a random 10% of the 1600 possible veteran records.Extracting: Three members of the research team (KMM, DL and DP) identified probable veterans in CRIS and extracted relevant data. Data fields were exported from CRIS and imported into a comma separated values file. Structured fields relating to the veterans’ socio-demographic characteristics, mental disorders and treatment pathways through care were extracted. The details were transferred to a bespoke study database. The 11 specific variables of interest extracted are shown in Table 1. The variables of interest were extracted through structured fields which are predefined within the CRIS system. In some cases, values were not present within the structured fields, this could be due to human or system error. Where possible, the research team sought to backfill missing data by reading through each patients’ clinical notes and identify the missing information.Analysing: Our utility and feasibility investigation used qualitative methods. This allowed us to discuss the practicality of the approach used. We focused on the utility and feasibility of identifying probable veterans and of extracting veteran data. In addition to the descriptive results, this paper also reports on the socio-demographic profiles, mental disorders and treatment pathways of the identified probable veterans.

### 2.3. Statistical Analysis

All analyses were undertaken using the statistical software package Stata (version 14.0, College Station, TX, USA). Statistical significance was defined as a *p*-value of less than 0.05. Percentages, frequencies and IQRs are presented, alongside chi-square or *t*-test statistics.

## 3. Results

### 3.1. Identifying

The identification process took place in two steps. First, a list of relevant military-related words was compiled by the research team, drawing on their extensive expertise of the AF. Second, similar, and frequently used, military-related terms that appeared in probable veterans’ records were derived during an initial search of veteran EHRs within CRIS. Together, these two methodologies resulted in a large list of military-related words, of which 19, including “Royal Navy”, “Army”, “Royal Air Force” and “Armed Forces” (see Table 2), proved to be useful. Using the same processes, we also created a list of over 30 exclusion limits, such as “navy blue” and “Salvation Army” (see Table 2), to prevent a substantial number of non-veterans being identified as veterans (false positives). The key words and exclusions were then combined, piloted and optimised, before being used to systematically search CRIS.

When considering the individual word searches used to identify probable veterans, the term “Army” returned the highest number of potential veteran records, followed by “Royal Navy” and then “Royal Air Force”. However, sheer numbers returned are not reflective of the hit rate of each term. We tested a random batch of 457 of the potential veteran records returned, to determine individual identification rates for each of the three key military terms used. “Royal Air Force” had the highest hit rate for correctly identifying veterans out of all 19 military-related words, followed by “Royal Navy” and then “Army” (see Figure 1).

The key word searches, combined with the exclusion limits, returned 6039 potential veterans. The research team randomly selected 1600 of these probable veterans to inspect in more detail, considering time and manpower restrictions. This sample size was determined based on the maximum time allowable for the research team to review each potential veteran. After a member of the research team (KMM, DL or DP) manually scrutinised each of these 1600 EHR records, 693 were identified as being probable veterans and were included in the final sample.

Identifying veterans through CRIS in this way was labour intensive and time consuming. Manually verifying each potential veteran record involved reading through, often, a vast number of free-text notes, written by the clinician who had seen the patient. We manually worked through 1600 probable veteran records for this project, which equated to 400 h of reading time or approximately 11 weeks’ worth of work. This time frame did not include creating the search strategies for identifying potential veterans, extracting and cleaning their data or carrying out data analyses.

### 3.2. Extracting

Following the identification of veterans, we were able to successfully extract data on 693 probable veterans who had accessed secondary mental health care services through the SLaM (see Table 3). However, the data available for extraction and analysis within CRIS was dependent on clinicians entering the information gathered from patients who had accessed care. Often EHRs contain large amounts of missing data—and this was true for several socio-demographic variables within the current study. Even after backfilling, 13.99% of probable veterans did not have a documented note of who they lived with in the database and 3.60% were missing details relating to their ethnicity. Backfilling equated to an estimated 60 h of reading time, to identify and fill data points of interest.

The sample had a median age of 74 years (IQR = 53–86) and the majority were male (90.76%). Probable veterans of both NS era and post NS era groups reported living alone most frequently (37.95%). The most common disorders diagnosed overall were depressive disorders (22.87%) and alcohol misuse disorders (10.46%). The sample had booked a median of 2 (IQR = 1–5), and attended a median of 2 (IQR = 1–4), outpatient secondary mental health care appointments through SLaM. The median number of inpatient secondary mental health care stays within SLaM across the full sample was 1 (IQR = 1–2) and the median duration of these inpatient stays was 28.5 days (IQR = 12–73.5).

The most common diagnosis for NS era probable veterans was depressive disorders and then anxiety disorders. The most common diagnosis for post NS era veterans was depressive disorders and then alcohol misuse disorders. A total of 233 (33.62%) veterans had a diagnosis unrelated to mental health. Differences were observed in the duration of inpatient mental health care stays, with NS era veterans experiencing a median stay of 60.5 days (IQR = 31–120) and post NS era veterans experiencing a median stay of 15 days (IQR = 7–35).

## 4. Discussion

This research showed that it is feasible to identify two groups of probable veterans who accessed secondary mental health care services through SLaM using CRIS. The hit rate for correctly detecting veterans from these EHRs was 43.3% and our final sample consisted of 693 veterans. We were able to identify probable veterans from pre and post NS era, but the procedure for doing so was far from straightforward. This study is the first to identify differences in mental disorder diagnoses and treatment duration between NS era veterans and post NS era veterans. Further, as far as we are aware, it is also the first to identify veterans and non-veterans accessing the same secondary mental health service. This study identified differences in prevalence rates between pre and post NS era and healthcare utilisation. This could be due to older veterans having more complex healthcare needs coupled with higher levels of comorbidity resulting in longer and more frequent inpatient stays.

An important strength of the study was the exploitation of EHRs, which are advantageous for investigating subgroups within the wider population. For example, we found that NS era veterans had longer inpatient stays within SLaM than post NS era veterans. Future research is required to investigate the drivers behind this observation and why differences may exist between two groups of individuals that had the same occupation. Another strength was that we tested the utility and feasibility of identifying UK veterans accessing NHS secondary mental health care for the first time. The study was unique in describing and evaluating how to use secondary health care records to undertake research on veteran’s secondary mental health care utilisation.

It must be noted that there was no way to confirm our identified CRIS veterans, verified by the research team using clinical notes, were *true* veterans; nor can we be confident of the integrity of the underlying primary data source. However, this is also the case for other EHR datasets, such as the Adult Psychiatric Morbidity Survey (APMS); for cohort studies, which require completers to self-report their veteran status [24]; and for hospitals in general. We were careful and deliberate about who to classify as a probable veteran in this study—we read through all clinical notes at least twice and only confirmed veteran status when an explicit statement about the patient serving was reported. Even so, the process relied both on patients self-reporting having previously served in the military—which may have been inaccurate, particularly considering this population of individuals were suffering from severe and complex mental health problems; as well as clinicians documenting veteran status in their notes. Further, applying the exclusion terms may have resulted in *false* negatives (i.e., wrongly excluding probable veterans), but it was not possible to evaluate this due to the large volumes of data used. It is worth noting, however, that protocol requires clinicians to talk through a patient’s previous occupations when they first enter mental health care services [25].

Along the same lines, there was no way to confirm that our verified non-veterans had not served in the AF. Some veterans may not have volunteered the fact that they had belonged to the military to their health care provider. In this case, there would be no mention of these individuals’ veteran statuses in their clinical notes. We can acknowledge that certain demographic and military factors, such as age and time since leaving the service, might impact on disclosure [26]—perhaps older veterans who left the military many years ago are less likely to reveal their history of serving in the AF than younger veterans who are still adjusting back to civilian life.

While we have shown that the process is feasible, the manual identification of probable veterans from CRIS was labour and resource intensive and time consuming. We recommend trying to improve the accuracy and efficiency of identifying veterans from EHR databases, such as CRIS, where possible. For example, as is already the case in NHS Scotland [27], the implementation of a military marker across the UK, perhaps one that could be verified with the Ministry of Defence’s records, would be extremely helpful. This would clearly indicate which patients had previously served in the AF, eliminating the reliance on self-reported veteran status and speeding up the manual identification process as a result. However, this approach could only identify newly presenting veterans and would not help with determining serving status retrospectively.

A more immediate solution to accelerating veteran identification is the creation of digitalised tools, such as natural language processing (NLP) methods, to automatically detect these individuals using keywords and rules. Of great importance is its utility in being applied automatically to EHR and free-text clinical notes [28,29]. NLP sub-themes, such as text mining, are represented as a set of programmatic rules or machine learning algorithms (i.e. automated learning from gold standard labelled data) to extract meaning from “naturally-occurring” text (meaning human generated text) [9,28]. The result is often an output that can be interpreted by humans with relative ease [30,31]. Previous work has developed successful NLP approaches for use within CRIS to identify subgroups of interest—for example, to allow the identification of patient suicide attempts [28,32], or linking education, social care and EHR records for adolescents mental health analyses with is the subject of ongoing work [33]. Considering this study, the creation of a similar NLP tool would ensure a consistent, reliable and effective approach to identifying veterans from free-text clinical notes. To the best of our knowledge, there is currently no tool within the UK that identifies veterans through secondary mental health care records in this way.

Because of time and resource constraints, we were only able to include 693 probable veterans. While this sample is large enough to show proof of concept, it is lacking in statistical power for more complex analyses, such as those involving subgroups of women or ethnic minority veterans. We have already detected approximately 6000 potential veterans from our current research. Based on this work, we expect that 30–40% could be verified as actual veterans using NLP, suggesting an anticipated sample size of 2000 veterans could easily be reached from CRIS. It would also be possible to match this veteran group to non-veterans accessing mental health services through SLaM. Using these data, we could establish whether there are similarities in socio-demographic, mental health, medication, suicide and treatment characteristics present between veterans and non-veterans, as well as between NS and post NS era veterans.

Data within EHRs are not collected primarily for research purposes and therefore often have large amounts of missing values [34]—as was the case here. While our included data can adequately test feasibility, such missing values decrease reliability and robustness. When time and manpower allow, we recommend backfilling missing data for outcome variables going forwards, to ensure data completeness. We could use participants’ clinical written notes to do this, by manually working through each patient’s records one-by-one. Details left out of the database’s structured fields are often included within these free-text fields [35], which would allow researchers to improve data quality. However, it must be acknowledged that these notes could be out-of-date and may introduce bias. Alternatively, we could use other known data sources to build a more comprehensive picture of the socio-demographic information of veterans seeking mental health treatment. We are aware that these additional data focus on patients who access primary mental health services, whose treatment needs are less complex than those who access secondary NHS care. However, some of our sample will have records in both systems, which would allow the exploitation of additional information for those individuals that do.

## 5. Conclusions

This study showed that it is feasible, although time consuming, to identify veterans accessing NHS secondary mental health care in the UK using CRIS and our manual approach. Our feasibility study contributes to the growing base of knowledge with regards to the mental health and needs of veterans and informs the development of digitalised approaches. Further, it is the first to identify differences between NS era and post NS era veterans. Despite our success in the current study, difficulties with the methodological procedure for identifying veterans have recognised a need for further work—including creating an automated NLP tool for detecting veterans more efficiently; increasing the pool of participants derived from CRIS; and addressing missing data.

## Figures and Tables

**Figure 1 healthcare-08-00001-f001:**
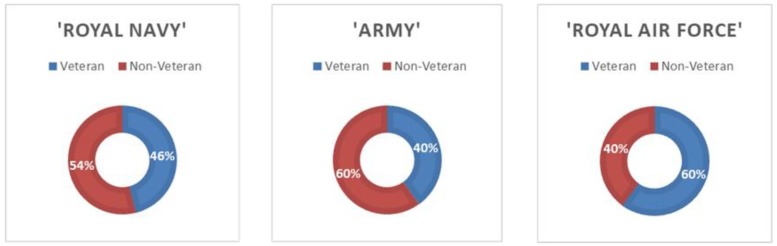
Hit rates for the three primary military search terms used in Clinical Record Interactive Search system—“Royal Navy”, “Army” and “Royal Air Force”.

**Table 1 healthcare-08-00001-t001:** Eleven variables extracted from the Clinical Record Interactive Search system.

Variables Extracted	
1. Age (in years) in 2018	7. Number of mental disorder diagnoses
2. Gender	8. Number of outpatient secondary mental health care appointments booked
3. Living arrangements	9. Number of outpatient secondary mental health care appointments attended
4. Ethnicity	10. Number of inpatient secondary mental health care stays
5. Age at mental disorder diagnoses	11. Duration of inpatient secondary mental health care stays (in days)
6. Types of mental disorder diagnoses	

**Table 2 healthcare-08-00001-t002:** Included key words, exclusion criteria and descriptive notes for the search terms used to identify veterans within the Clinical Record Interactive Search system.

Included Key Words	Exclusion Criteria	Notes
Army	“who was/is in (the) army”	Majority of times this refers to someone other than the patient
	“Salvation Army”	
	“army knife”	
	“army gear”	
	“army style”	
	“army cadet”	
	“army cadette”	
	“army themed”	
	“child army”	
	“army family”	
	“rebel army”	
	“refugee army”	
	“army service”	
	“private army”	
	“army green”	
	“army <item of clothing>”	Clothing
	“army type”	
	Foreign armies: Eritrea, Sri Lanka	Reference to service in non-UK army, or experiences relating to non-UK army
Navy	“navy blue”	Clothing
	“dark navy”	Clothing
	“navy colour”	Clothing
	“wearing (a) navy”	Clothing
	“dressed in navy”	Clothing
	“navy <item of clothing>”	Clothing
	“Merchant Navy”	
	“Army and Navy Store”	
	“worked for Navy, Army, Air Force Institute”	NAAFI
	“<family member> was/is in (the) navy”	Family member in Navy
	“due to join the Navy”	(Thinking of) joining Navy
	“accepted into Navy”	(Thinking of) joining Navy
	“potential careers, including Navy”	(Thinking of) joining Navy
	Foreign navies: Italian, US, Israeli, Portuguese, Burmese, Eritrea	Reference to service in non-UK navy, or experiences relating to non-UK navy
RAF/air force	“<family member> was/is in (the) RAF”	Family member in RAF
Armed Forces		
Afghan		Deployment location
Iraq		Deployment location
Bosnia		Deployment location
Kosovo		Deployment location
Falklands		Deployment location
N Ireland		Deployment location
Cyprus		Deployment location
Germany		Deployment location
Enlisted		
National service		
Veteran		
Combat Stress		Military charity
SSAFA		Military charity
Help for Heroes		Military charity

**Table 3 healthcare-08-00001-t003:** Descriptive statistics for the 693 identified probable veterans.

	Overall (*n* = 693)	*n* Missing (*n*, %)	NS Era (*n* = 349)	*n* Missing (*n*, %)	Post NS Era (*n* = 344)	*n* Missing (*n*, %)	Chi^2^ (*p*)	*p* Value
Age at sampling point (years; 2018) [median, IQR]	74 (53–86)	-	86 (82–90)	-	52 (41–61)	-	-	<0.001
Gender (*n*, %) -Male -Female	629 (90.76) 64 (9.24)	-	317 (90.83) 32 (9.17)	-	312 (90.70) 32 (9.30)	-	0.0037 (0.952)	-
Residency (*n*, %) -Alone -Friends/family/other -Partner/children	263 (37.95) 108 (15.58) 225 (32.46)	97 (13.99)	129 (36.96) 39 (11.17) 147 (42.12)	34 (9.74)	134 (38.95) 69 (20.05) 78 (22.67)	63 (18.31)	50.614 (<0.001)	-
Ethnicity (*n*, %) -White British -Other	610 (88.02) 58 (8.36)	25 (3.60)	330 (94.55) 15 (4.29)	4 (1.14)	280 (81.39) 43 (12.50)	21 (6.10)	21.215 (<0.001)	-
Number of veterans with an inpatient admission (*n*, %)	146 (21.07)	-	59 (16.90)	-	87 (25.29)	-	-	0.068
Number of veterans with an outpatient appointment (*n*, %)	116 (16.74)	-	35 (10.02)	-	81 (23.54)	-	-	<0.001
Number of veterans with an inpatient admission and outpatient appointment (*n*, %)	36 (5.19)	-	16 (4.58)	-	20 (5.81)	-	-	0.162
Age at mental disorder diagnosis (years) [median, IQR]	71 (46–83)	52 (7.50)	82 (77–87)	11 (3.15)	46 (36–55)	41 (11.91)	-	<0.001
Types of mental disorder diagnoses (*n*, %) -Alcohol use disorders -Drug disorders -Stress disorders -Depressive disorders -Anxiety disorders -Schizophrenic disorders -Personality disorders -Other mental disorders *	*n* = 822 86 (10.46) 34 (4.13) 75 (9.12) 188 (22.87) 62 (7.54) 40 (4.86) 37 (4.50) 67 (8.15)	-	*n* = 306 16 (5.22) 1 (0.32) 24 (7.84) 88 (28.75) 27 (8.82) 15 (4.90) 5 (1.63) 9 (2.94)	-	*n* = 516 70 (13.56) 33 (6.39) 51 (9.88) 100 (19.37) 35 (6.78) 25 (4.84) 32 (6.20) 58 (11.24)	-	39.607 (<0.001) 32.160 (<0.001) 11.341 (0.001) 1.302 (0.254) 1.264 (0.261) 2.808 (0.094) 21.228 (<0.001) 40.460 (<0.001)	-
Number of comorbid mental health diagnoses (*n*, %) -Zero -One -Two -Three or more	336 (48.48) 190 (21.41) 76 (10.96) 26 (3.75)	65 (9.37)	164 (46.99) 122 (34.95) 40 (11.46) 11 (3.15)	12 (3.43)	172 (50.00) 68 (19.76) 36 (10.46) 15 (4.36)	53 (15.40)	13.927 (0.016)	-
Number of outpatient appointments booked ** [median, IQR] *** -Booked -Attended	2 (1–5) 2 (1–4)	-	2 (1–3) 1.5 (1–3)	-	3 (1–13) 2 (1–9)	-	-	0.293 0.186
Number of inpatient mental health care stays [median, IQR] ***	1 (1–2)	-	1 (1–2)	-	1 (1–2)	-	-	0.124
Duration of inpatient mental health care stays (in days) [median, IQR] ***	28.5 (12–73.5)	-	60.5 (31–120)	-	15 (7–35)	-	-	0.038

* Other mental disorders include dementia, dissociative disorders, somatoform disorders, eating disorders, sexual disorders, developmental disorders, hyperkinetic disorders, self-poisoning, self-harm, tic disorders and intellectual disabilities; ** Appointments booked for management or treatment of a mental health diagnosis; *** Patients who had zero values were labelled as missing.

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
