# Peer review of "Identifying Veterans Using Electronic Health Records in the United Kingdom: A Feasibility Study"

_healthcare, 2019, doi:10.3390/healthcare8010001_

Round 1

Reviewer 1 Report

The reviewer strongly recommend this work to be reconsider after minor revision on journal.
However, some comment should pay attention to improve the quality of paper:

1. This paper focus on a feasibility study for identifying the veterans with mental health problems and without problems.
To evaluate the method of proposed mechanism with CRIS, authors should present how to extract variables as shown in  table 1 in 2.2.1 study procedure.

2. Authors depicted the paper as report form, to improve the quality of paper, introduction of NLP algorithm and additional method should be described in discussion chapter 4.

Author Response

Comment: The reviewer strongly recommend this work to be reconsider after minor revision on journal. However, some comment should pay attention to improve the quality of paper:

Response: We are grateful for the due diligence of the Reviewer and have addressed there comments hereafter.

Comment: This paper focus on a feasibility study for identifying the veterans with mental health problems and without problems. To evaluate the method of proposed mechanism with CRIS, authors should present how to extract variables as shown in  table 1 in 2.2.1 study procedure.

Response: We have amended the manuscript (line 155-156) to reflect that the variables included in this work are from structured fields pre-defined within CRIS.

Comment: Authors depicted the paper as report form, to improve the quality of paper, introduction of NLP algorithm and additional method should be described in discussion chapter 4.

Response: We have amended the manuscript to include a brief discussion of the utility of NLP. Please see line 292-296.

Reviewer 2 Report

This is an interesting manuscript which reads well. It does require some revision.

The introduction has summarised the literature however there is not much information on the theoretical consideration in the Methods section. Perhaps the authors can consider highlighting the theory they have drawn from-the only mention of any theory is in the discussion section. Additionally, the definitions of NS and post NS veterans could be included in the study population description (methods section) and not in the study procedure section. The authors should consider including all the study prodcures in one section and not include some in the results section. Only the results should be presented in this section. What are the references for the Natural Language Processing (NLP) on line 285? Perhaps the authors could provide some reasons why the findings are different for the two groups of veterans. Why did they decide to group the veterans into two? Please jusify why this was necessary. Could the inpatient stays for the NS be longer due to older age/comorbidities?

Author Response

Comment: The introduction has summarised the literature however there is not much information on the theoretical consideration in the Methods section. Perhaps the authors can consider highlighting the theory they have drawn from-the only mention of any theory is in the discussion section.

Response: We are grateful to the Reviewer for their helpful comments. We have amended the manuscript to provide further clarification on the theoretical approach employed in this study. Several changes have been made to the manuscript and are tracked using track changes.

Comment: Additionally, the definitions of NS and post NS veterans could be included in the study population description (methods section) and not in the study procedure section.

Response: We have amended the manuscript to include the statement about National Service era veterans into the study population.

Comment: The authors should consider including all the study prodcures in one section and not include some in the results section.

Response: We thank the Reviewer for their comments and appreciate the suggestion on placing the study procedures into one place. We feel it is more suited in the result section, while being descriptive, it does describe how we arrived at our keywords; which is part of the findings of this study.

Comment: What are the references for the Natural Language Processing (NLP) on line 285?

Response: Based on other Reviewer comments, we have amended the manuscript to provide further details about the Natural Language Processing.

Comment: Perhaps the authors could provide some reasons why the findings are different for the two groups of veterans.

Response: We have amended the manuscript to include a brief commentary on differences.

Comment: Why did they decide to group the veterans into two? Please jusify why this was necessary.

Response: We have amended the manuscript to further clarify this point. Please see line 138-142.

Comment: Could the inpatient stays for the NS be longer due to older age/comorbidities?

Response: We believe this is the case for the National Service era group but further research is required. We have amended the manuscript to reflect this.